

# Determinants of age-related decline in walking speed in older women

Valéria Feijó Martins[1,*], Luigi Tesio[2,3,*], Anna Simone[2], Andréa Kruger Gonçalves[1] and Leonardo A. Peyré-Tartaruga[1]

[1] LaBiodin Biodynamics Laboratory, Universidade Federal do Rio Grande do Sul, Porto Alegre, Rio Grande do Sul, Brazil
[2] Department of Neurorehabilitation Sciences, IRCCS Istituto Auxologico Italiano, Milan, Italy
[3] Department of Biomedical Sciences for Health, University of Milan, Milan, Italy
* These authors contributed equally to this work.

Corresponding author
Leonardo A. Peyré-Tartaruga,
leonardo.tartaruga@ufrgs.br

## ABSTRACT

**Background:** Walking speed is reduced with aging. However, it is not certain whether the reduced walking speed is associated with physical and coordination fitness. This study explores the physical and coordination determinants of the walking speed decline in older women.

**Methods:** One-hundred-eighty-seven active older women (72.2 ± 6.8 years) were asked to perform a 10-m walk test (self-selected and maximal walking speed) and a battery of the Senior fitness test: lower body strength, lower body flexibility, agility/ dynamic balance, and aerobic endurance. Two parameters characterized the walking performance: closeness to the modeled speed minimizing the energetic cost per unit distance (locomotor rehabilitation index, LRI), and the ratio of step length to step cadence (walk ratio, WR). For dependent variables (self-selected and maximal walking speeds), a recursive partitioning algorithm (classification and regression tree) was adopted, highlighting interactions across all the independent variables.

**Results:** Participants were aged from 60 to 88 years, and their self-selected and maximal speeds declined by 22% and 26% ($p < 0.05$), respectively. Similarly, all physical fitness variables worsened with aging (muscle strength: 33%; flexibility: 0 to −8 cm; balance: 22%; aerobic endurance: 12%; all $p < 0.050$). The predictors of maximal walking speed were only WR and balance. No meaningful predictions could be made using LRI and WR as dependent variables.

**Discussion:** The results suggest that at self-selected speed, the decrease in speed itself is sufficient to compensate for the age-related decline in the motor functions tested; by contrast, lowering the WR is required at maximal speed, presumably to prevent imbalance. Therefore, any excessive lowering of LRI and WR indicates loss of homeostasis of walking mechanics and invites diagnostic investigation.

## INTRODUCTION

Walking speed is frequently investigated in the older adult population (*Mian et al., 2006*; *Schoene et al., 2017*; *Frimenko, Goodyear & Bruening, 2015*). Aging, even under healthy conditions (*Michel & Sadana, 2017*), is associated with visible stiffness in ambulation, more prudent walking, and quantitative changes in virtually all walking parameters. Such

changes include shorter stride length and frequency (hence, lower speed), larger step width, reduced trunk mobility, and increased risk of falls (*Mian et al., 2006*; *Aboutorabi et al., 2016*; *Herssens et al., 2018*; *Schoene et al., 2017*). Spontaneous walking speeds below 1.0 m s$^{-1}$ are associated with increased mortality (*Cesari et al., 2005*; *Figgins et al., 2021*). Reduced walking speed is also related to metabolic/cardiovascular, and mental/neurologic comorbidities (*Mone & Pansini, 2020*; *Mone et al., 2022a*, *2022b*; *Noh et al., 2020*; *Zanardi et al., 2021*).

Declines in the most diverse body functions can contribute to changes in walking performance (*Cruz-Jimenez, 2017*; *Miller, Bemben & Bemben, 2021*; *Pantoja et al., 2016*). Cardiorespiratory endurance reaches its maximum capacity at about 20 years of age, after that, until 65 years of age, there is a 20–30% reduction in cardiac output (*Erkkola, Vasankari & Erkkola, 2021*). Maximal muscle strength reduces by 15–30% every 10 years after the fifth decade of life (*Papadopoulou, 2020*; *Pantoja et al., 2016*). Muscle power production may also decrease because of mitochondrial dysfunction (*Conley et al., 2007*). Changes in joint flexibility may explain the lower range of joint excursions, subtended mainly by loss of joint cartilage and a decrease in collagen concentration, entailing loss of compliance and elasticity of joint capsules, ligaments, and tendons (*Kothari et al., 2016*; *Erkkola, Vasankari & Erkkola, 2021*). Decreased balance seems to induce changes in walking speed (*Cruz-Jimenez, 2017*) as the decrease in body balance may stem from delayed muscle recruitment; impaired anticipatory and compensatory postural adjustments; loss of proprioceptive fibers (*Sanders et al., 2019*; *Gerards et al., 2021*; *Martina et al., 1998*); and decreased stiffness of calf tendons, leading to delayed elongation of muscle spindles (*Onambele, Narici & Maganaris, 2006*). The age-associated decline in static and dynamic balance variables related to postural sway has been estimated at 1% per year (*Takeshima et al., 2014*).

Changes in neural control also play an important role in age-related changes in walking mechanics (*Mian et al., 2006*; *Ortega & Farley, 2007*). Furthermore, neural control can provide functional compensation for metabolic and dynamic losses due to, *e.g.*, muscle overactivation and co-activation (*Ortega & Farley, 2007*; *Miller, Bemben & Bemben, 2021*; *Delabastita et al., 2021*). A simple, more general form of adaptation is lowering the walking speed. However, this adaptation is not without disadvantages. The muscular work during walking is minimized by an inverted pendulum (*Alexander, 2005*). Maximizing the effectiveness of this mechanism requires a given speed (*Cavagna, Thys & Zamboni, 1976*) and, for any given speed, a given step length and, therefore, a given cadence (*Cavagna & Franzetti, 1986*). The optimal step length, at optimal speed, is very close to those spontaneously adopted by young adults (*Cavagna, Thys & Zamboni, 1976*; *Peyré-Tartaruga & Monteiro, 2016*). Lower or higher speeds imply higher external work and cost metabolic (*Tesio, Roi & Möller, 1991*). For any given speed, increasing cadence implies a higher muscular work to reset, at each step, the limbs with respect to the body center of mass (*Willems, Cavagna & Heglund, 1995*).

Studies assessing spontaneous walking speed in older adults have obtained contradictory results that seem highly sample-dependent (*Herssens et al., 2018*; *Fukuchi, Fukuchi & Duarte, 2019*; *Boulifard, Ayers & Verghese, 2019*). Speed measures range from

0.79 (*Boulifard, Ayers & Verghese, 2019*) to 1.34 m s$^{-1}$ (*Fukuchi, Fukuchi & Duarte, 2019*). Naturally, speed needs to be normalized by body height (or lower limb length), *e.g.*, through the dimensionless Froude number (*Cavagna & Franzetti, 1986*). Another possible determinant of reduced preferred speed is dynapenia (reduced muscle strength). This reduction would imply an impairment in the forward propulsive function of the gastrocnemius muscle (*Conway et al., 2021*).

The comparison of possible mechanisms of reduced speed between older women and men is lacking. Walking habits (*Leung et al., 2009*) and body size can interact with age in determining walking speed and cadence. Even though the reduction in walking speed is similar between the sexes, women reduce stride length proportionally more than men, reducing stride frequency less than men (*Frimenko, Goodyear & Bruening, 2015*).

Whether walking speed depends on physical fitness and why healthy older adults tend to adopt slower speeds even over short distances is still an open question. A previous study has revealed that step time (inverse of step frequency) had the greatest influence on the reduction of walking speed in senior women (*Fien et al., 2019*). Thus, some coordination parameters, such as walk ratio index (WR) and balance, seem to be related to reduced speed due to the task of swinging limbs during walking (*Gomeñuka et al., 2020*). Also, a recent retrospective cohort study has reported that long-term participation in a community-based exercise program delays age-related declines in walking speed and lower extremity muscle strength (*Hayashi et al., 2021*); however, other physical fitness parameters were not evaluated.

This study aims to investigate the motor skills that determine the walking speed in older women. Factors that affect step length and frequency, such as muscle strength and balance, are candidates to explain the reduced functional mobility of older women.

## MATERIALS AND METHODS

This is an open, cross-sectional study carried out in a university extension program in southern Brazil. The study was approved by the local ethics committee (Universidade Federal do Rio Grande do Sul, project number: 17243819.0.0000.5347 and clinical trials ID: NCT04348539). All individuals who agreed to participate signed an informed consent form. For the Istituto Auxologico Italiano, this study fell within the RESET research program, Ricerca Corrente IRCCS, Italian Ministry of Health.

### Participants

One hundred eighty-seven untrained older women were recruited through the media (including social media). All were non-frailty individuals (Fried frailty index).
The recruitment was on the School of Physical Education, Physiotherapy and Dance website of the Federal University of Rio Grande do Sul (https://www.ufrgs.br/esefid/site/). The site has a space for disseminating studies to the community. Inclusion criteria for sample selection were age between 60 and 90 years, community-dwelling status, regular physical training program in the last 3 months at least two sessions per week, and verbal understanding instructions for testing and demonstrating independent ambulation. Exclusion criteria included the use of assistive mobility devices or any walking limitation.
## Assessments instruments

For the measurement of self-selected walking speed (SSWS), the 10-m walk test was used. And the participants were asked to walk three attempts at their preferred usual speed in a 14-m straight line, measured at 10-m and discarded the first and last 2 m, which correspond to the period of acceleration and deceleration of the walking (*Novaes, Miranda & Dourado, 2011*). The same procedure was applied to determine the maximal walking speed. Here, the participants were instructed "to walk as fast as possible without running". The time, in seconds, was measured using a digital stopwatch, and the mean of three repetitions was used for further analysis. Speeds are presented in meters per second.

The locomotor rehabilitation index (LRI) was calculated as the ratio of the observed walking speed to the predicted optimal (lowest cost) walking speed (*Peyré-Tartaruga & Monteiro, 2016*; *Gomeñuka et al., 2019*). Subject's optimal walking speed was estimated using the dimensionless Froude number (Fr), as shown in Eq. (1):

$$Fr = v^2/(g \times L) \tag{1}$$

where v is the speed, g is the gravity acceleration, and L is the lower limb length (measured from the anterior-inferior iliac spine to the ground through the lateral malleolus) (*Vaughan & O'Malley, 2005*). The dimensionless optimal walking speed (OWS, Eq. (2)) in humans corresponds to Fr = 0.25 (Eq. (1)). So that,

$$OWS = \sqrt{0.25 \times g \times L} \tag{2}$$

Thus, the LRI is as follows (Eq. (3)):

$$LRI = 100 \times SSWS/OWS \tag{3}$$

The LRI has been applied to assess different populations, including patients with heart failure (*Figueiredo et al., 2013*), Parkinson's disease (*Monteiro et al., 2017*), and older adults trained in Nordic walking (*Gomeñuka et al., 2019*).

The WR was calculated as the ratio of step length to cadence (*Sekiya & Nagasaki, 1998*; *Rota et al., 2011*; *Bogen et al., 2018*; *Kalron et al., 2020*), with step length expressed in mm and cadence in steps $min^{-1}$. The WR serves as a sensitive indicator of neural and cognitive walking impairments: it significantly decreases in multiple sclerosis (*Rota et al., 2011*; *Kalron et al., 2020*) and Parkinson's disease (*Zanardi et al., 2021*) as well as in healthy subjects under high attentional demands (*Almarwani et al., 2019*).

Four tests were used to assess motor parameters that potentially influence walking mechanics. Tests are from the Senior Fitness Test battery (*Rikli & Jones, 1999*) (see legend of Table 1 for short descriptions): (i) 8-foot up and go (agility/dynamic balance test, ABa), (ii) 30-s chair stand (lower body strength, LBS), (iii) 2-min step (aerobic endurance, AE), and (iv) chair sit and reach (lower body flexibility, LBF). These tests have been extensively validated, do not require any special equipment, and can be easily applied in any clinical or exercise environment (*Rikli & Jones, 2013*; *Gonçalves et al., 2021*).

## Statistical analysis

The predictive models for either SSWS or maximal walking speed and either LRI or WR were applied. Given that multicollinearity is expected across variables describing a subject's motor performance (mostly between maximal walking speed and SSWS but also between speed and LRI), a decision-tree model rather than a conventional multiple regression model was used.

SSWS, maximal walking speed, LRI, and WR data were tested for normality of distribution based on skewness and kurtosis and then summarized as mean (standard deviation, SD) and median (interquartile range, IQR) or median (IQR) when appropriate. Significance was set at $p < 0.05$, and p-values were Bonferroni-adjusted for multiple comparisons. A predictive regression model was applied using a recursive partitioning algorithm, *i.e.*, a classification and regression tree (CART) model. This analysis is distribution-free and transforms continuous levels into ordinal grades. The algorithm builds a decision tree based on binary splits on variables (either continuous, ordinal, or categorical). At each split, nodes are generated, and these nodes can be further split. The algorithm automatically detects interactions (*i.e.*, the tree/node structure) between independent variables, providing the highest explanation of variance for the dependent variable (either categorical or continuous; here, continuous). The final result (terminal nodes) comprises a series of classes with the lowest possible within-class variance and the highest possible between-class variance. Unlike conventional linear regression modeling, in which the analyst must specify the expected interactions, CART itself discovers interactions, even high-order ones that are very difficult to hypothesize (*Breiman et al., 1984*). The algorithm is more sensitive to interactions than to main effects. The model's variance explanation is much less vulnerable to multicollinearity issues. Each split is performed on a single variable. The latter is ignored if no further information is added by further splitting on a covariate. Software packages typically allow the analyst to control the procedure by imposing a minimum number of observations on each node or by setting stopping rules for tree branching (for a simple clinical example, see *D'Alisa et al., 2006*). *A priori* knowledge or requirements can thus complement the purely algebraic search for the maximum amount of variance explained. The stability of the predicted model can be inferred either by imposing the model splits (from the building sample) to an independent (validation) sample or by simulating several independent samples (boot-strapping) originating from the available sample. This procedure is typically done through random extraction of subsamples and substituting their values by random replication of observations from the remaining sample or the original total sample (resampling). In any case, the amount of variance explained for the validated tree unavoidably declines (shrinks) concerning the variance explained for the original sample. It is left to the analyst to decide whether the model is satisfactorily stable or not (*Breiman et al., 1984*). There is no rule of thumb for accepting a given amount of variance explained. A reasonable empirical threshold for the validation tree is 30%, as suggested by the results for trees effectively predicting the length of stay, care costs, and functional outcomes of rehabilitation inpatients in the USA (*Stineman, 1995*).

**Table 1 Descriptive statistics for the variables in the study.**

|  | Mean (SD) | Median (IQR) | Range |
|---|---|---|---|
| **Age** (years) | 72.22 (6.8) | 72 (67–77) | 60–88 |
| **Height** (m) | 1.56 (0.06) | 1.56 (1.53–1.61) | 1.39–1.73 |
| **BMI** | 28.37 (4.67) | 27.95 (25.22–30.88) | 19.85–49.07 |
| **LRI** (%) | 90.0 (13.83) | 90.5 (80.6–100.4) | 60.1–120.7 |
| **WR** | – | 0.56 (0.52–0.63) | 0.35–1.02 |
| **LBS** (no. full stands) | – | 16 (13–19) | 6–30 |
| **LBF** (cm)* | −3.44 (10.70) | −2 (−9 to 3) | −29 to 25 |
| **ABa** (seconds)* | – | 5.1 (4.52–5.65) | 3.37–8.85 |
| **AE** | 87.50 (15.89) | 87 (79–97) | 53–128 |
| **SSWS** (m s$^{-1}$) | 1.30 (0.22) | 1.31 (1.16–1.42) | 0.77–1.87 |
| **MWS** (m s$^{-1}$) | 1.74 (0.30) | 1.74 (1.57–1.90) | 0.94–2.74 |

Notes:
* The lower the value, the better the condition.
m, meters; BMI, body mass index (mass height$^{-2}$); LRI, locomotor rehabilitation index; WR, walk ratio; no., number; LBS, lower body strength (number of full stands in 30 s with arms folded across chest); LBF, lower body flexibility [from sitting position at front of chair, with leg extended and hands reaching toward toes, number of cm (+ or −) from extended fingers to tip of toe; negative values: cm missing to toes contact]; cm, centimeter; ABa, agility/dynamic balance test (number of seconds required to get up from seated position, walk 8 foot, turn, and return to seated position on chair); AE, aerobic endurance (number of full steps completed in 2 min, raising each knee to point midway between patella and iliac crest-score is number of times right knee reaches target); SSWS, self-selected walking speed; MWS, maximum walking speed.

In the present study, CART analysis is initiated from unsplit dependent variables (SSWS, maximal walking speed, LRI, WR) (root nodes). Each variable was split into nodes according to optimal cut-off points for the remaining variables to maximize the variance explained. Splitting continued until terminal nodes were defined, building the final classification model. The limitations imposed on each tree were as follows: maximum splitting levels, 10; splitting algorithm, least squares; minimum size node to split, 10; minimum rows allowed in a node, 5; tree pruning and validation method, cross-validation; the number of cross-validation folds, 10; and tree pruning criterion, within one standard error of minimum cost complexity.

Descriptive statistics and regression modeling were done using IBM SPSS® version 21.0 (IBM Corporation, Armonk, NY, USA), and STATA® software (version 16.0; Stata Corp. LLC, College Station, TX, USA). CART analysis was done through DTREG® software (DTREG, Brentwood, TN, USA, 2021).

# RESULTS

Table 1 provides descriptive statistics and a short definition of all variables assessed in this study.

Age, SSWS, maximal walking speed, LRI, WR, and the four independent variables (ABa, LBS, AE, and LBF) were tested for normality based on skewness and kurtosis (Bonferroni-adjusted $p < 0.006$). Only ABa, LBS, and WR were significantly nonnormal (data not shown); thus, the assumption of linear regression was violated. For each of these variables, observations smaller than or greater than three SDs beyond the mean were trimmed (for linear regression only, not for further analyses). The WR ratio remained nonnormal

**Table 2 Linear regression modeling.**

| | n | β (95% CI) | Const (95% CI) | R² | p[#] | Change[$] |
|---|---|---|---|---|---|---|
| **LRI** | 185[*] | −0.419 [−0.711 to −0.128] | 120.2 [99.08–141.3] | 0.04 | 0.0051 | −14% |
| **WR** | 187 | −0.003 [−0.005 to −0.001] | 0.803 [0.641–0.966] | 0.04 | 0.0087 | −15% |
| **LBS** | 185[^] | −0.158 [−0.245 to −0.070] | 27.30 [20.95–33.66] | 0.06 | 0.0005 | −33% |
| **LBF** | 187 | −0.331 [−0.555 to −0.108] | 20.49 [4.282–36.70] | 0.04 | 0.0039 | 107%[&] |
| **ABa** | 183[^] | 0.048 [0.030–0.065] | 1.718 [0.461–2.976] | 0.14 | 0.0000 | 22% |
| **AE** | 187 | −0.364 [−0.700 to −0.029] | 113.8 [89.50–138.1] | 0.02 | 0.0333 | −12% |
| **SSWS** | 187 | −0.009 [−0.133 to −0.004] | 1.932 [1.600–2.266] | 0.07 | 0.0002 | −22% |
| **MWS** | 187 | −0.014 [−0.020 to −0.008] | 2.726 [2.283–3.168] | 0.10 | 0.0000 | −26% |

Notes:
[#] Bonferroni adjusted significance level, 0.006.
[*] Two missing data for LRI.
[^] Observations exceeding the mean by ±3SD were trimmed.
[$] Percent change from predicted values at 60 and 88 years; Positive changes indicate worsening for ABa and LBF; negative changes indicate worsening for LBS and AE.
[&] From 0 to −8 cm.
Dependent variable (age) versus locomotor rehabilitation index (LRI), walk ratio (WR), lower body strength (LBS), lower body flexibility (LBF), agility/dynamic balance test (ABa), aerobic endurance (AE), self-selected walking speed (SSWS) and maximal walking speed (MSW β, slope coefficient of linear regression; const, y-intercept of linear regression; CI, confidence interval; R², proportion of variance explained.

($p < 0.005$ for skewness and kurtosis), as it had a uniform distribution. Linear regression was applied despite this limitation.

Table 2 revealed that all variables worsened with age (with confidence limits never including zero). The change was not significant for AE. In any case, the worsening was moderate. From 60 to 88 years of age, SSWS and maximal walking speed worsened (*i.e.*, declined) by 22% and 26%, respectively. For the other variables, worsening ranged from 14% to 33%. For LBF, a percentage change would be misleading: finger–toe distance increased from 0 to 8 cm. The variance explained by age was low for all variables, exceeding 10% for WR.

The correlation matrix of the nine variables (Fig. 1) gives an overview of bivariate associations.

Low values of AE (cardiorespiratory fitness index) and LBF (joint flexibility index) indicate better performance. Fig. 1 shows that most of Pearson's correlation coefficients were very low. Only the correlation coefficients between LRI and SSWS (0.96), LRI and maximal walking speed (0.51), and SSWS and maximal walking speed (0.53) were higher than the arbitrary threshold of |0.5|. These findings were expected (see Eqs. (1)–(3)), given that these variables are either derived from each other (LRI and maximal walking speed or SSWS) or strictly dependent on the subject's height (maximal walking speed and SSWS).

Interactions between multiple variables were explored through CART analysis. Fig. 2 depicts the decision trees used to predict SSWS (left panel) and maximal walking speed (right panel).

Figure 3 shows the trees developed to predict LRI (left panel) and WR (right panel).

Table 3 summarizes the variance explained (for training/building and validation data) for each of the four trees shown in Figs. 2 and 3.

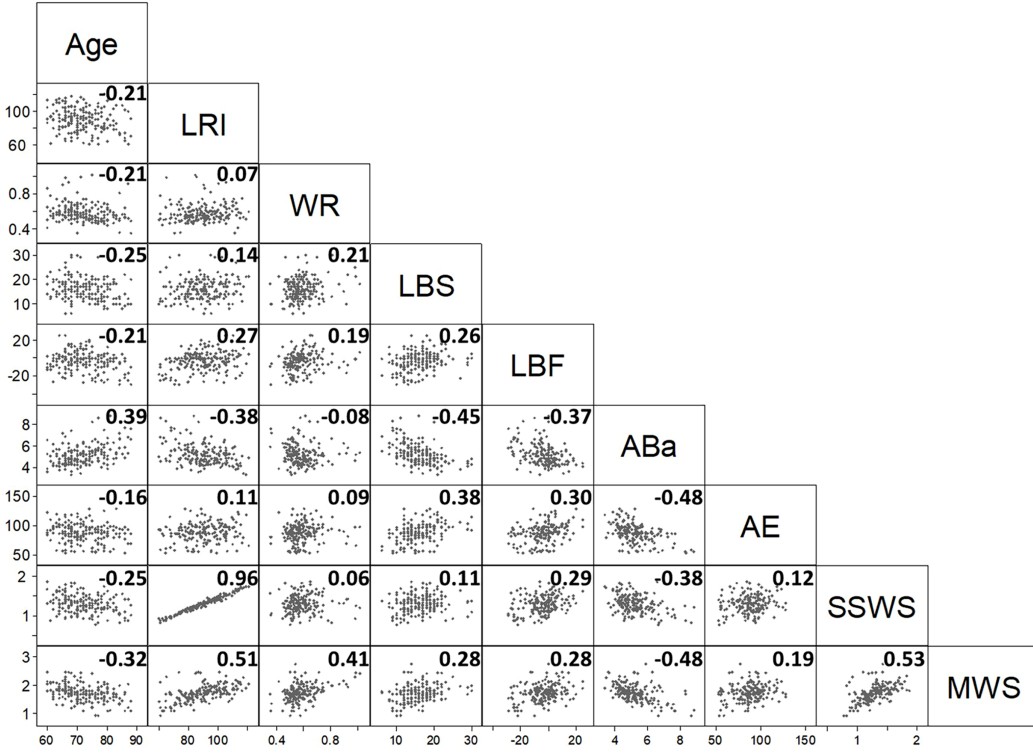

**Figure 1 The scatterplot provides the correlation half-matrix of the parameters. Pearson's correlation coefficients are given in the corresponding boxes.** Parameters: age, locomotor rehabilitation index (LRI), walk ratio (WR), lower body strength (LBS), lower body flexibility (LBF), agility/dynamic balance test (ABa), aerobic endurance (AE), self-selected walking speed (SSWS), and maximal walking speed (MSW).

As demonstrated in Table 3, the variance explained by validation trees was satisfactory for SSWS, maximal walking speed, and LRI (ranging from 36% to 93%) but barely acceptable for WR (21%). The results suggest that most independent variables, including age, were not predictive of SSWS. In the corresponding tree, only LRI was retained, a circular finding (see above). By contrast, maximal walking speed was explained by SSWS (another expected finding) and, notably, by WR for speeds below 1.23 m s$^{-1}$ as well as by ABa for WR values of less than or equal to 0.7 (most of the cases).

## DISCUSSION

The expected associations between SWSS and maximal walking speed did not convey meaningful information. Although expected, the association between SWSS and LRI indicates that 7% of the variance in SWSS is related to size effects. Therefore, LRI seems to be an improved marker of functional mobility due to size-dependent variation in walking speed. Other points deserve consideration. Neither age nor any of the four motor indices selected (LBS, LBF, Aba, and AE, see legend of Table 1) nor WR explained SSWS. Maximal walking speed was partially explained by the interaction between WR and ABa (Fig. 2). The WR tree (Fig. 3) confirms the relationship of WR with speed.

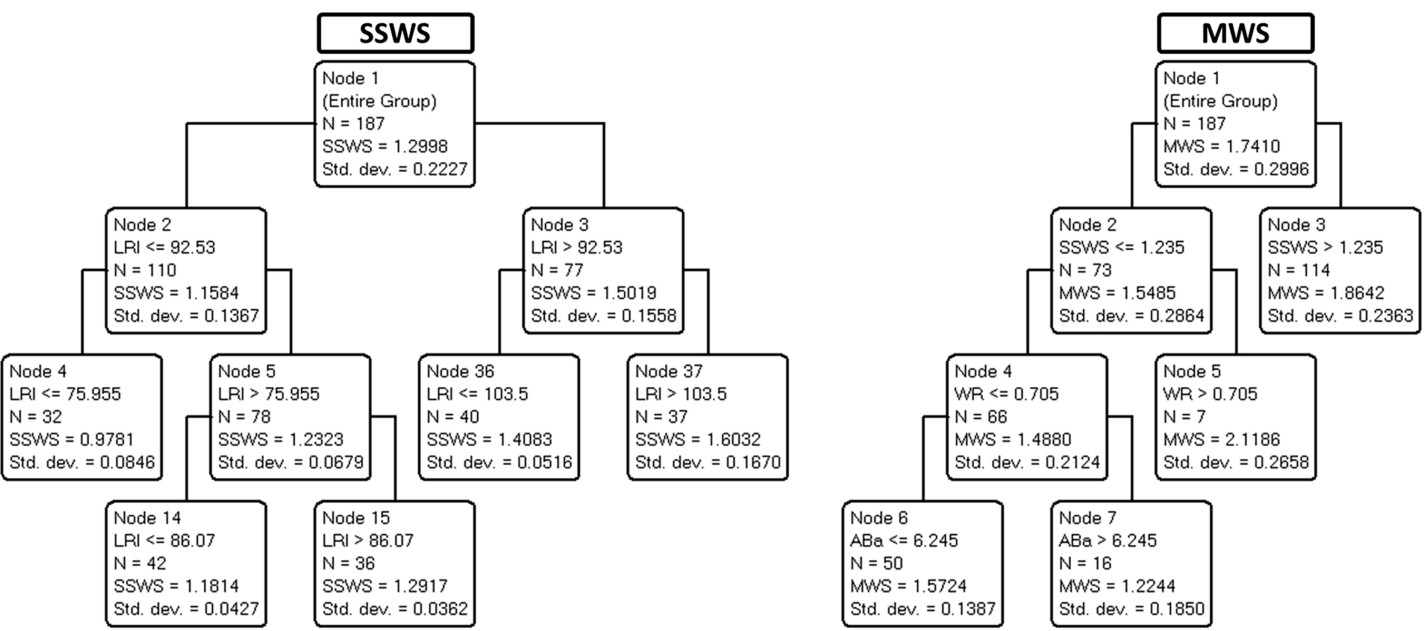

**Figure 2 The scatterplot provides the correlation half-matrix of the parameters. Pearson's correlation coefficients are given in the corresponding boxes.** Parameters: age, locomotor rehabilitation index (LRI), walk ratio (WR), lower body strength (LBS), lower body flexibility (LBF), agility/dynamic balance test (ABa), aerobic endurance (AE), self-selected walking speed (SSWS), and maximal walking speed (MWS).

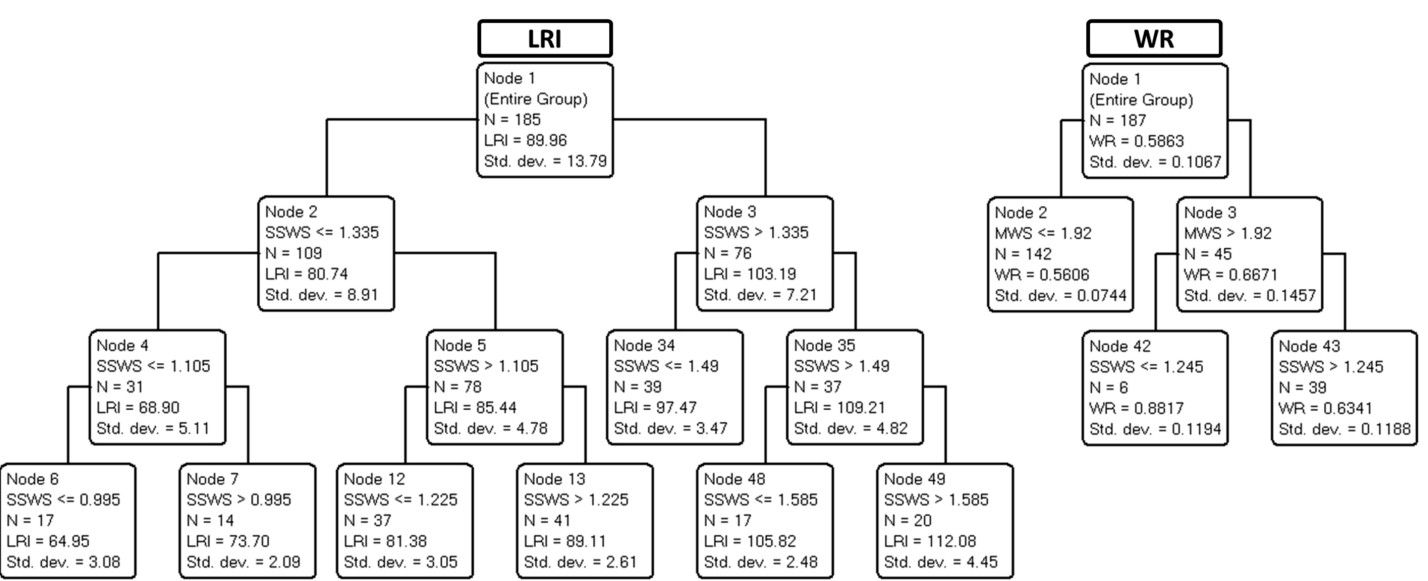

**Figure 3 Final classification and regression tree (CART) prediction models of locomotor rehabilitation index (LRI) and walk ratio (WR).** Parameters: age, locomotor rehabilitation index (LRI), walk ratio (WR), lower body strength (LBS), lower body flexibility (LBF), agility/dynamic balance test (ABa), aerobic endurance (AE), self-selected walking speed (SSWS), and maximal walking speed (MWS).

**Table 3 Variance explanation of the decision trees (Figs. 2 and 3) for the locomotor rehabilitation index (LRI), walk ratio (WR), self-selected walking speed (SSWS) and maximal walking speed (MSW).**

| Target variable | Variance explanation % | |
|---|---|---|
| | Training data | Validation data |
| LRI | 95% | 93% |
| WR | 33% | 21% |
| SSWS | 84% | 80% |
| MWS | 50% | 36% |

## An algebraic explanation

It must be said that the explanation of variance requires some variance to be explained and a covariance. Along a 28-year gradient, the spontaneous and maximal speeds of older women undergo small changes, with a high interindividual variation. On the other hand, WR is largely invariant with speed and age. Not surprisingly, the weak relationship between speed and age (Table 1) is lost if a bivariate association is abandoned in favor of an interactive model (Figs. 2 and 3). Of course, a greater sample size might have allowed obtaining a more branched and explanatory decision tree. This algebraic interpretation, however, does not seem to be entirely satisfactory. An interpretation based on physiology from outside the data should be considered based on numerical assumptions.

## Looking for an explanation in physiology

The results suggest that healthy aging implies a mild tendency for a decrease in SSWS, unexpectedly unrelated to the various physical performance parameters analyzed and the step length/cadence ratio (WR). The question then arises: given that these women were capable, at various ages, of increasing their speed (on average, SSWS was 1.30 m s$^{-1}$, whereas maximal walking speed was 1.74 m s$^{-1}$), why did they not retain the same SSWS at all ages? The second unanswered question is, how could WR remain unrelated to age and the various motor performance parameters? After all, step length and cadence should reflect lower limb joint power, mobility, and balance.

One reason may be that human walking has very wide margins of safety. In symmetric gaits, overall energy expenditure is minimal, given the refined pendulum-like exchange of mechanical energy of the center of mass. This characteristic makes humans the most efficient walkers in the animal realm (*Sockol, Raichlen & Pontzer, 2007*; *Henn, Cavalli-Sforza & Feldman, 2012*). The cardiorespiratory power and the power required to drive muscles (mainly the plantar flexors) remain much below the ceiling level (*Tesio et al., 2017*). Lower limb joint excursions retain wide mobility margins despite having a more overall flexed posture. In focal strength deficits, compensation may occur between limbs and, within the same limb, between joints (*Tesio, Roi & Möller, 1991*; *Tesio & Rota, 2019*). Once the speed needs to be decreased (see below for further explanation), there seems to be no need for taking longer and more frequent steps than that already foreseen for the new speed. In case of need, however, a wide margin of safety remains for decreasing WR.

In fact, at any given speed, a decrease in step length has a minimal influence on the effectiveness of the pendulum mechanism until a 50% decrease is reached (*Cavagna & Franzetti, 1986*; *Tesio et al., 2017*).

Therefore, as a form of speculation, it can be hypothesized that cardiac–energetic or musculoskeletal constraints do not determine the age-related decline in speed. Rather, as suggested by several authors (*Ortega & Farley, 2007*; *Miller, Bemben & Bemben, 2021*; *Delabastita et al., 2021*), balance control may represent a hidden, relevant determinant of the mild age-related decrease in SSWS.

## The role of balance compared to other walking constraints

Over short distances, one can well afford a mildly higher metabolic cost unless this is prevented by severe cardiac or respiratory deficits (*Tesio, Roi & Möller, 1991*; *Willems, Cavagna & Heglund, 1995*). However, because of its pendulum-like mechanics, the body center of mass must be accelerated forward, upward (*Cavagna, Thys & Zamboni, 1976*), and laterally (*Tesio & Rota, 2019*) at each step to overcoming ground friction and gravity acceleration; the greater the ground friction, the longer the step, and the faster the movement, the shorter the step duration. These mechanical demands decrease by reducing walking speed. In particular, such a decrease in speed leaves more time for the amazingly fast U-turn from one side to the opposite at each step, as demonstrated by the analysis of the 3D trajectory of the body center of mass during a single stance (*Malloggi et al., 2021*).

Once the speed is conveniently lowered, therefore, a further decrease in step length (as evidenced by a lower WR) would unnecessarily entail a higher "internal" work per unit distance, *i.e.*, the muscular work needed to reset the limbs at each step (*Willems, Cavagna & Heglund, 1995*). Not surprisingly, WR remained nearly invariant with age and SSWS in the present sample of women, confirming literature data on a wide range of velocities and adult ages (*Rota et al., 2011*; *Bogen et al., 2018*). This invariance, however, does not hold for the maximal walking speed showing that the spatiotemporal coordination pattern represented by WR in the present study is altered in aged women at high walking speeds.

Consistently with its explanatory role, balance is known to decrease in healthy aging. In the present study, ABa was the variable that most depended on age (Table 2). It entered the prediction algorithm of maximal walking speed together with WR only. These results point toward a pivotal role of balance in determining the decline of speed in aging, at least at higher speeds. It should be noted that WR is diminished whenever the balance is primarily affected (see Introduction). At any speed, WR decreases when walking on slippery surfaces (*Cappellini et al., 2010*), and, as a rule, in the case of neural impairments. Furthermore, the higher co-activation of lower limb muscles (*Mian et al., 2006*; *Gomeñuka et al., 2020*) may help to understand balance's role in reducing walking speed in older women. The typical WR for adults and older adults up to 85 years is in the order of 5.5–6.5 mm step$^{-1}$ min$^{-1}$, across a wide range of walking speeds and body heights (*Sekiya & Nagasaki, 1998*; *Rota et al., 2011*), and in line with our findings. Of note, this parameter is consistently lower by about 5% in women than men (*Bogen et al., 2018*).

### Aging and walking, and what LRI and WR tell us

To sum up, in healthy aging, the decrease in speed (either self-selected or maximal) is modest (*Sanders et al., 2019*; *Gerards et al., 2021*; *Martina et al., 1998*). LRI and WR, which are related to each other, are virtually stable and seem unrelated to cardiorespiratory and musculoskeletal performance. This result is no surprise, given the high effectiveness of human bipedalism. LRI and WR indices provide complementary information. They both seem to reflect a homeostatic control of walking, so alterations might represent alarming early predictors of latent cardiorespiratory or joint power limitations (LRI) and/or latent balance deficit (LRI and WR). In particular, a decrease in LRI indicates a reduced pendulum-like mechanism resulting in a higher energy cost of walking (reduced economy, *Gomeñuka et al., 2014*, *2016*; *Peyré-Tartaruga & Monteiro, 2016*).

Further, a reduced WR may indicate balance deficits insufficiently compensated for by a reduction of speed. In support of this speculation, one should consider that human bipedalism is unique among bipedal vertebrates in many respects. For instance, the role of plantar flexion is critical as the main "engine" of walking (*Usherwood et al., 2012*). Another unique feature of particular interest is the need for a refined balance control on the frontal plane (*Malloggi et al., 2021*; *Cassidy et al., 2014*). This need can represent a weakness in the case of balance deficits and many other neural impairments, leading to a reduction in speed and, in the most severe cases, further reduction of step length.

Some limitations of the present study cannot be overlooked. First, the results refer only to women. Second, only short distances were tested; speed and LRI and WR indices might have differed at longer distances. Third, the sample size did not validate the predictive model in an independent sample, representing a complementary and perhaps a more robust mode of validation than cross-validation. Finally, questions regarding the chosen statistical methods may have some implications for the results. Future studies in this field are advised including and controlling factors as comorbities and including groups more advanced with symptoms of frailty as in institutionalized individuals (*Mone & Pansini, 2020*; *Mone et al., 2022a*, *2022b*).

## CONCLUSIONS

The maximal walking speed was partially explained by an impaired agility/dynamic balance, and a reduction in muscle strength, flexibility, and balance across age groups was observed. Whereas LRI seems to denote physical capabilities, WR represents a key coordination aspect of functional mobility, particularly related to balance in older women. The results suggest that both LRI and WR are helpful as a short screening battery for walking performance in aging and, potentially, in disability. These indices, however, only measure the presence of complex, tenacious, adaptive, and homeostatic mechanisms so that any alterations should entail a deeper, causal diagnostic inquiry.

## ACKNOWLEDGEMENTS

The authors acknowledge the participants for their voluntary involvement in this study. We extend our acknowledgements to all Brazilian citizens who allow so many researchers to improve their scientific knowledge in public graduate programs.

### Funding

This study was supported by Capes, Fapergs and CNPq (Capes-financing code 001, for the masters and doctorate scholarships, FAPERGS/MS/CNPq-08/2020-PPSUS-21/2551-0000094-5, and FAPERGS-PqG-2017-17/2551-0001038-8). The Istituto Auxologico Italiano-IRCCS, collaborated on this work within the RESET research project, Ricerca Corrente IRCCS, Ministero della Salute, Italy. The funders had no role in study design, data collection and analysis, decision to publish, or preparation of the manuscript.

### Grant Disclosures

The following grant information was disclosed by the authors:
Capes, Fapergs and CNPq: FAPERGS/MS/CNPq-08/2020-PPSUS-21/2551-0000094-5, and FAPERGS-PqG-2017-17/2551-0001038-8.
Istituto Auxologico Italiano-IRCCS.
RESET research project, Ricerca Corrente IRCCS, Ministero della Salute, Italy.

### Competing Interests

Leonardo A. Peyré-Tartaruga is an Academic Editor for PeerJ.

### Author Contributions

- Valéria Feijó Martins conceived and designed the experiments, performed the experiments, analyzed the data, prepared figures and/or tables, authored or reviewed drafts of the article, and approved the final draft.
- Luigi Tesio analyzed the data, prepared figures and/or tables, authored or reviewed drafts of the article, and approved the final draft.
- Anna Simone analyzed the data, prepared figures and/or tables, authored or reviewed drafts of the article, and approved the final draft.
- Andréa Kruger Gonçalves conceived and designed the experiments, performed the experiments, analyzed the data, authored or reviewed drafts of the article, and approved the final draft.
- Leonardo A. Peyré-Tartaruga conceived and designed the experiments, performed the experiments, analyzed the data, authored or reviewed drafts of the article, and approved the final draft.

### Human Ethics

The following information was supplied relating to ethical approvals (*i.e.*, approving body and any reference numbers):

The Universidade Federal do Rio Grande do Sul granted Ethical approval to carry out the study within its facilities (project number: 17243819.0.0000.5347).

### Data Availability

The raw measurements are available in the Supplemental File.

## Supplemental Information

Supplemental information for this article can be found online at http://dx.doi.org/10.7717/peerj.14728#supplemental-information.

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
