# Peer review of "Determinants of age-related decline in walking speed in older women"

_PeerJ, doi:10.7717/peerj.14728_

## Round 0.1 · original submission · Major Revisions

The three reviewers and I see much potential in this paper but also some areas for further improvement prior to this being considered for publication in PeerJ. I would like to see a little bit more detail in the introduction about previous studies that have looked to predict gait speed in older adults using similar gait data to your study. The following study would be one such example https://pubmed.ncbi.nlm.nih.gov/29200084/

Reviewer 1 has suggested that you cite specific references. You are welcome to add it/them if you believe they are relevant. However, you are not required to include these citations, and if you do not include them, this will not influence my decision.

Reviewer 1 ·

Basic reporting

The manuscript should be revised by a fluent English speaker. Also, The discussion should focus on the role of comorbidities in frailty. Please, cite and discuss:
- Gait Speed Test and Cognitive Decline in Frail Women With Acute Myocardial Infarction.
Mone P, Pansini A.
Am J Med Sci. 2020 Nov;360(5):484-488. doi: 10.1016/j.amjms.2020.03.021. Epub 2020 Apr 14.
PMID: 32409104
- Correlation of physical and cognitive impairment in diabetic and hypertensive frail older adults.
Mone P, Gambardella J, Lombardi A, Pansini A, De Gennaro S, Leo AL, Famiglietti M, Marro A, Morgante M, Frullone S, De Luca A, Santulli G.
Cardiovasc Diabetol. 2022 Jan 19;21(1):10. doi: 10.1186/s12933-021-01442-z.
PMID: 35045834
- Outcomes of hospitalized patients with COVID-19 according to level of frailty.
Andrés-Esteban EM, Quintana-Diaz M, Ramírez-Cervantes KL, Benayas-Peña I, Silva-Obregón A, Magallón-Botaya R, Santolalla-Arnedo I, Juárez-Vela R, Gea-Caballero V.
PeerJ. 2021 Apr 13;9:e11260. doi: 10.7717/peerj.11260. eCollection 2021.
PMID: 33954054
- Frailty severity is significantly associated with electrocardiographic QRS duration in chronic dialysis patients.
Chao CT, Huang JW.
PeerJ. 2015 Oct 22;3:e1354. doi: 10.7717/peerj.1354. eCollection 2015.
PMID: 26528415
- Physical decline and cognitive impairment in frail hypertensive elders during COVID-19.
Mone P, Pansini A, Frullone S, de Donato A, Buonincontri V, De Blasiis P, Marro A, Morgante M, De Luca A, Santulli G.
Eur J Intern Med. 2022 May;99:89-92. doi: 10.1016/j.ejim.2022.03.012. Epub 2022 Mar 14.
PMID: 35300886
- Frailty and Risk of Fractures in Patients With Type 2 Diabetes.
Li G, Prior JC, Leslie WD, Thabane L, Papaioannou A, Josse RG, Kaiser SM, Kovacs CS, Anastassiades T, Towheed T, Davison KS, Levine M, Goltzman D, Adachi JD; CaMos Research Group.
Diabetes Care. 2019 Apr;42(4):507-513. doi: 10.2337/dc18-1965. Epub 2019 Jan 28.
PMID: 30692240

Experimental design

Ok

Validity of the findings

Ok

Additional comments

Nothing

Reviewer 2 ·

Basic reporting

Some points have not been fully understood. The experimental procedure needs to be better explained.
If possible, improve the quality of figures. The manuscript should be revised by a fluent English speaker.
Please, revise the paper for minor spell check.

Experimental design

This work is of significant interest but there are some critical issues. For example, to give greater relevance to this work, the aspect of comorbidities could be included.
Recently, articles have been published that analyze these aspects.
I recommend viewing the following articles:
doi: 10.1016/j.ejim.2022.03.012
doi: 10.1186/s12933-021-01442-z

Validity of the findings

The findings are interesting but need to be better discussed with the prevously suggested manuscripts.

Reviewer 3 ·

Basic reporting

This manuscript requires major revisions to strengthen the study, more references are required to support your statements throughout the introduction and discussion to strengthen your manuscript.

Throughout the manuscript you interchange between walking speed and gait speed, I would suggest one and be consistent throughout. Likewise with maximal conditions and maximal walking speeds.

In terms of numbering, where you have a number under 10, recommend writing out the number, e.g., 2 = two. In regards to starting sentences, I recommend not starting with an abbreviation eg. line 170.

I also suggest the removal of "We" in your manuscript".

There is inconsistency with the reporting of your references within the reference list, recommend a review.

Experimental design

The experimental design aligns with the aims and scope of the journal. However, the inconsistency of the terminology used concerning walking and gait and the two conditions the participants are measured against are not as clear and concise as they should be.

Within the abstract, the background information is missing, currently, there is only the aim, need to provide 1-2 sentences on the topic itself.

In materials and methods, more information is required as to where the participants were recruited from location-wise and what was required within the "suitability for a complete assessment. I also recommend adding in some information about cognitive screening, given the age of the participant. you see to include/exclude over physical but what about cognitive and the ability to understand and follow instructions?

In regards to the testing, how was the walking/gait speed measured - you seem to be missing that part, was it by stopwatch?

Results section, you will need to include more information than just table 1 provides descriptive, how many in total did you recruit from and what % did you recruit, how many males/females, mean and SD age.

Validity of the findings

The study is meaningful and once major revisions have been applied will be strengthened for the audience to show the impact and value of the study.

Table 1 requires acronyms/abbreviations to be spelled out in the note section underneath. For consistency, the values represented in Tables 1 and 2 need to be consistent where applicable for the decimal places.
Table 3 heading - target variable is not required to be all in capital letters.

The conclusion needs to be tighter - more clear and more concise is required to link to the aim and results of the study.

Additional comments

An attachment has been provided.

Annotated reviews are not available for download in order to protect the identity of reviewers who chose to remain anonymous.

---

## Round 0.2 · Minor Revisions

The reviewers and I thank you for your amendments so far. Please look at the remaining requests from Reviewer 1 prior to this manuscript being suitable for acceptance in PeerJ.

Reviewer 3 ·

Basic reporting

Comments have been addressed from the first review. An attachment has been provided on the minor comments to be addressed and included.

Experimental design

Comments have been addressed from the first review. An attachment has been provided on the minor comments to be addressed and included.

Validity of the findings

Comments have been addressed from the first review. An attachment has been provided on the minor comments to be addressed and included.

Additional comments

Comments have been addressed from the first review. An attachment has been provided on the minor comments to be addressed and included.

Annotated reviews are not available for download in order to protect the identity of reviewers who chose to remain anonymous.

---

## Round 0.3 · accepted · Accept

Thanks for your hard work in addressing the reviewers comments. We are happy to accept this manuscript for publication in PeerJ.

Reviewer 3 ·

Basic reporting

no comment

Experimental design

no comment

Validity of the findings

no comment

Additional comments

great work with this manuscript. I have no further comments or suggestions to be made.